# Effect of Dosing Interval on Compliance of Osteoporosis Patients on Bisphosphonate Therapy: Observational Study Using Nationwide Insurance Claims Data

**DOI:** 10.3390/jcm10194350

**Published:** 2021-09-24

**Authors:** Hyunil Lee, Sangcheol Lee, Dokyung Kim, Weonmin Cho, Sungtan Cho, Siyeong Yoon, Soonchul Lee

**Affiliations:** 1Department of Orthopaedic Surgery, Ilsan Paik Hospital, Inje University, Goyang 10380, Korea; lhi76@naver.com (H.L.); I0497@PAIK.AC.KR (S.C.); 2CHA Bundang Medical Center, Department of Orthopaedic Surgery, CHA University School of Medicine, Pocheon-si 13488, Korea; scheln@naver.com (S.L.); fkzhfjqm@naver.com (D.K.); science5019@naver.com (W.C.)

**Keywords:** osteoporosis, bisphosphonate, compliance, big data

## Abstract

Only a few studies are available on the effect of the dosing interval of bisphosphonate on drug compliance. We analyzed the data of patients who were newly prescribed bisphosphonate using a national insurance claims database. Drug compliance was assessed by calculating medication possession ratio (MPR) over a minimum of a 1-year follow-up. This analysis included 281,996 new bisphosphonate users with a mean age of 68.9 years (92% women). The patients were divided into daily, weekly, monthly, 3-monthly, and switch groups (who changed the drug to other dosing intervals). The average MPR was the highest in the switch group (66%), and the longer the dosing interval, the higher the compliance (3-monthly, 56% vs. daily, 37%). “Non-compliant” was defined as an MPR under 80%. Various factors which were possibly associated with “non-compliant” MPR were investigated using multiple regression analysis. Multivariate analysis showed that male patients were more likely to be non-compliant with pharmacotherapy than female patients, with as odds ratio of 1.389. Younger patients had a significantly lower likelihood of being non-compliant than older patients for age 60–69 vs. age 80+. Long dosing intervals were recommended to improve compliance and special attention was given to older and male patients.

## 1. Introduction

Osteoporosis is common in postmenopausal women and the number of affected individuals is expected to increase gradually as the population continues to age [1]. Patients with osteoporosis are susceptible to pathologic fracture, and fracture-related costs are likely to increase by over USD 22 in the United States billion by 2025 [2]. Similar to hypertension and diabetes mellitus, there is no specific symptom of osteoporosis until pathologic fracture occurs. Therefore, there is a low awareness of the necessity of taking an anti-osteoporosis drug. According to studies, only 24–40% of the patients received pharmacological therapy [3,4]. To reduce the risk of osteoporotic fracture, it might be important to make high-risk patients take appropriate drugs, as well as to continue drug therapy for a sufficient period. Compliance with osteoporosis treatment is reported to be correlated with an improvement in bone mineral density (BMD) and the subsequent reduction in fracture risk [5]. Nevertheless, drug compliance is still poor and two out of five people are known to discontinue the drug in the first year [6]. Other studies suggest that only half of the patients continue bisphosphonate therapy for 1 year, and 43% for 1 to 2 years, owing to various reasons [7].

Bisphosphonates account for about 70% of all anti-osteoporosis drugs [8]. Oral bisphosphonates are known for adverse gastrointestinal effects and, to avoid this, bisphosphonates should be taken on an empty stomach with a large amount of water and the patients should remain in an upright position for 30 min. To reduce the inconvenience of taking medication and improve compliance, pharmaceutical companies developed higher doses of bisphosphonates which could be taken at less frequent intervals than the current forms (weekly, monthly, or even yearly). The extension of the dosing interval to a annual regimen was found to improve gastrointestinal safety, and showed an increased persistence and the same effectiveness [9,10]. There are several studies comparing the compliance with bisphosphonate therapy at various dosing intervals; however, the results are still controversial. Moreover, few studies deal with the compliance according to dosing interval using the nationwide insurance claims database in South Korea. Given that our population is continuously aging, and the burden of osteoporotic fracture is rapidly growing, identifying the dosing interval that has a high compliance is crucial. A detailed understanding of factors affecting the continued medication experience may improve compliance and thereby reduce fracture. In the current study, we measured and compared the compliance to bisphosphonate therapy with various dosing intervals by checking the medication possession ratio (MPR). Our hypothesis was that “There would be a better compliance of drug in patients taking bisphosphonate with the longer dosing interval.” This study investigated if the dose interval was associated with drug compliance and the patient-related factors affecting this compliance.

## 2. Materials and Methods

### 2.1. Data Source

We conducted an observational study using administrative claims data in South Korea. Administrative claims data was initially collected for insurance purposes but was highly valuable for research. The study population was chosen from the National Health Insurance Service (NHIS) database, including information regarding insurance eligibility, medical treatment, health examination, and medical care institution. NHIS covered 98% of the population of Korea. Medical treatment data consisted of electronic bills for the medical treatment provided, prescription of drugs, and diagnosis codes (as defined by the International Classification of Diseases, 10th revision), as well as treatment costs for claims. Therefore, almost all information about patients and diseases could be obtained from the Korean NHIS database, and this database was used in many epidemiological studies [11,12,13,14]. This study was conducted according to the guidelines of the Declaration of Helsinki. Informed consent was waived, and the study design was approved by the ethics review board of Bundang Cha hospital (protocol code 2018-10-024 and date of approval: 2 May 2019). This study was reported in accordance with STROBE guidelines.

### 2.2. Study Population

We selected patients over the age of 50 years, diagnosed with osteoporosis and prescribed bisphosphonates between 01 January 2016 and 31 December 2016. This period was set as the index period (Figure 1). Using the NHIS database, patients with osteoporosis were categorized under the following disease codes: M80 (osteoporosis with pathological fracture), M81 (osteoporosis without pathological fracture), or M82 (osteoporosis in diseases classified elsewhere). The prescription of anti-osteoporosis drugs was limited to bisphosphonate or bisphosphonate complex. Both oral and intravenous forms of bisphosphonate were included. We did not investigate the use of other classes of anti-osteoporosis drugs such as selective estrogen receptor modulator or parathyroid hormone. We included only patients whose data were available until 31 December 2017 for a minimum of a 1-year follow-up. The exclusion criteria were as follows: (1) We tracked the data for 1 year from the index period (between 01 January 2015 and 31 December 2015) to evaluate the prescription of the anti-osteoporosis drug. The data from this period were used as wash-out data to exclude the patients who were prescribed bisphosphonates before the index period. History of anti-osteoporosis drugs in one year prior to index date was evaluated for each patient. Through this exclusion, we tried to select the new users for bisphosphonates strategically. (2) We also excluded the patients who had underlying diseases, which could affect drug prescriptions for diseases such as cancer, dementia, or Paget’s disease. (3) Patients who died during the study period were excluded; (4) patients who took other classes of anti-osteoporosis drugs besides bisphosphonates, and (5) patients who took a yearly dose of bisphosphonates were also excluded owing to the relatively short follow-up period; 3.5% of the patient population took a yearly dose of bisphosphonate.

The flow-chart and number of patients in each stage is depicted in Figure 2. By selecting patients under these criteria, we were able to collect data on new users of bisphosphonate. The observation started from the index date (the day when patients first took prescription) until 31 December 2017. The minimum follow-up period was around 1 year and the maximum follow-up period for certain patients was around 2 years.

### 2.3. Demographic Data

Basic demographic data extracted from the database was as follows: age, sex, area of residence (metropolis, city, or rural area), insurance fee (top 0–25 percentile, 25–50 percentile, 50–75 percentile, or bottom 75–100 percentile), history of pathologic fracture (presence/absence), fracture site, history of surgery for pathologic fracture (presence/absence), and site of surgery. Age was categorized into four groups: 50–59, 60–69, 70–79, and 80 years and older (80+). Area of residence and insurance fee were utilized to estimate the effect of patient’s socioeconomic status on drug compliance. We investigated the common fracture sites such as hip, spine, humerus, wrist, or multiple sites in the wash-out and index period (from 01 January 2015 to 31 December 2016). Information on surgery was searched using the procedure code for fracture fixation of the spine/humerus, joint arthroplasty for the shoulder, fracture fixation of the femur, joint arthroplasty for the hip, and fracture fixation of the forearm bone. Bisphosphonate prescription was classified into one of five groups according to the purchase records: daily, weekly, monthly, 3-monthly, or switch group. The patients whose bisphosphonate dosing interval was changed during the follow-up period to another dosing interval by switching the drug were assigned to the “switch group”. Switching means the change of dosing interval regardless of the kind of active substance. Information about the currently available bisphosphonate was obtained from the Korea Pharmaceutical Information Center (http://www.health.kr/main.asp; accessed date 19/Oct/2020). The information for this specific type of bisphosphonate was provided in Appendix A.

### 2.4. Measure of Compliance

We defined compliance according to the International Society for Pharmacoeconomic and Outcomes Research [15,16]. Compliance was defined as “the extent to which a patient acts in accordance with the prescribed interval, and dose of a dosing regimen”. This “compliance” during the follow-up period was measured using the medication possession ratio (MPR). MPR was calculated as the days of prescription divided by the days of total follow-up duration and obtained as a percentage. For example, if a patient was prescribed three packets of weekly bisphosphonate (each packet containing four pills recommended to be taken once a week) for the total follow-up duration of 1 year, the MPR would be calculated as follows: (3 packets × 4 pills × 7 days)/365 × 100 = equal to 23%. “The compliant rate” or “proportion of compliant patients” was determined by calculating how many patients had an MPR of 80% or higher [17,18].

### 2.5. Statistical Analyses

We presented descriptive characteristics according to dosing interval groups. Categorical variables were compared using Chi-square test. The analysis of variance (ANOVA) test or t-test for drug compliance was performed according to patients’ demographics. Statistical significance was set at 5%. Patients with MPR of ≥80% were considered compliant, and simple comparison of various demographic factors between compliant and non-compliant patients was performed using the Chi-square test. Multivariate logistic regression analysis was performed to identify the factors affecting compliance while adjusting for confounding factors such as age, sex, fracture site, and economic status. The results were presented with odds ratios and 95% confidence intervals (CIs). We used the R software (version 3.2.4; R Foundation for Statistical Computing) for statistical analyses. In our study, the *p* value was regarded as unimportant because of the extremely large number of patients. The *p*-value could be reduced by increasing the sample size [19]. The *p*-value had limited value in large-scale data; therefore, rather than relying solely on the *p*-value, emphasis was placed on the difference between groups, the odds ratios of the groups or the range of confidence interval, depending on factors.

## 3. Additional Analyses with Longer Follow-Up

We performed an auxiliary study with longer follow-up data to overcome the limitation of short follow-up which was criticized by reviewer during review process. Because original database with large number of patients did not contain longer follow-up data, we used another dataset which was provided by the same agency. Patients’ selection processes and analyses were almost identical to the process of the original data, except that the wash-out period (2 year), index period (3 year), and follow-up period (5 year) were extended compared to the previous analysis (Appendix A). Briefly, the number of initially included patients was 34,674 from 01 January 2008 to 31 December 2010. After exclusion, 6455 new bisphosphonate users were finally selected (Appendix A). Their demographics and the pattern of dosing interval were analyzed in the same manner as the original dataset. They were followed up to 31 December 2015 to make a minimum 5-year follow up to calculate MPR.

## 4. Results

### 4.1. Demographics of the Patients

We analyzed 281,996 patients diagnosed with osteoporosis and newly prescribed with bisphosphonate from 01 January 2016 to 31 December 2016. The baseline characteristics of the study population were summarized according to dosing interval in Appendix A. Most of patients were women (92%) aged 60–69 (35%) and 70–79 (32%). The most prescribed dose was weekly (44%), followed by 3-monthly (24%), monthly (15%), and daily (2%). Sixteen percent of the patients changed to a different dosing interval in the follow-up period and were categorized as the “switch group” (Figure 3A); 11% of the patients sustained fractures during index period: Spine fracture (45.9%) was the most prevalent, followed by forearm (18.4%), hip (8.4%), and humerus fractures (3%). Additionally, 2.8% of patients underwent surgery for a pathological fracture during the index period.

Figure 3B show the dosing interval, stratified on age group. A similar trend was seen in all age groups. However, generally, older patients were likely to receive 3-monthly doses, and in contrast, weekly doses were commonly prescribed to younger patients. Male patients were more likely to receive weekly doses than female patients (81.9% vs. 40.4%, Appendix A). Patients from rural areas were most likely to receive 3-monthly doses or the switching of drugs than patients from cities, whose common dosage was weekly. The insurance fee did not seem to be associated with the pattern of dosing interval. Neither presence of the fracture nor the location of the fracture had an impact on the dosing interval, although patients with fractures had a slightly higher possibility of taking 3-monthly doses.

### 4.2. Drug Compliance

The mean MPR of all patients was 49.6% (±34 standard deviation (SD)). Regarding age, the patients in aged 60–69 showed the highest mean MPR of 52.7% and patients aged 80+ showed the lowest of 42.6% (Figure 4A and Appendix A). Mean MPR was significantly higher in women (50.7% vs. 38.3% for men, Figure 4B and Appendix A). The insurance fee was not meaningfully associated with a difference in MPR. Although some of the comparisons were statistically significant, the actual difference was within 1.3%. The overall compliance of patients from rural areas was lower than that of patients from cities, but the difference was small (Figure 4D and Appendix A). Surprisingly, the presence or absence of fracture or the history of surgery due to fracture did not correlate meaningfully with the overall MPR (Appendix A). Moreover, the MPRs did not differ considerably according to the fracture site and history of surgery for fracture. The MPR was slightly lower for patients who underwent surgery for a forearm fracture and slightly higher for patients who performed surgery for multiple fractures (Figure 4F and Appendix A).

The highest MPR according to dosing interval was reported for the switch group as 65.7% (±28.3, Appendix A and Figure 5). The second highest MPR was reported for the 3-monthly group as 56.4%. The lowest MPR was reported for the daily dose group as 37.1%. Long dosing intervals generally showed enhanced MPR. MPR according to each follow-up period is shown in Figure 6. Drug compliance (MPR) gradually decreased, irrespective of the dosing intervals in all groups.

Our original data had a short follow-up period of a minimum of 1 year, so long-interval dosing had the better starting point. To overcome this potential bias, we conducted a minimum 5-year follow-up study with another dataset. Data showed that the MPR increased as the dosing interval increased, just as in the 1-year follow-up data (Figure 7).

### 4.3. Factors Affecting Compliance Rate

Overall, 28% of patients were compliant, with an MPR of more than 80%. In other words, 72% of the subjects were not compliant with bisphosphonates (MPR < 80%). The percentages of compliant patients were 21%, 20%, 28%, and 34% for the daily, weekly, monthly, and 3-monthly doses, respectively (Table 1). The compliance rate of the switch group was the highest at 41%.

The results of the univariate analysis of the effects of various factors on the compliance rate are presented in Table 1. The results showed that the compliant and non-compliant patients differed in all examined factors including sex, insurance fee, area of residence, dosing interval, and history of fracture and surgery. After the adjustment for covariates by multivariate logistic regression analysis, the odds ratio of a patient being non-compliant with treatment was 46% lower among 3-monthly-dose users than among daily dose users (odds ratio [OR] 0.54, 95% CI 0.501–0.582, Table 2). The adjusted odds ratio of being non-compliant was 39% higher in males than in females (OR 1.39, 95% CI 1.34–1.44). Patients in younger age groups (aged from 50 to 79) showed lower odds ratio of being non-compliant (0.565–0.674) than patients aged 80+ (Table 2). Patients who had a fracture (OR 0.95, 95% CI 0.900–0.996) or history of surgery for a fracture (OR 0.885, 95% CI 0.856–0.914) were more likely to continue using the drug, but the odds ratio was not high (11% and 5% less non-compliant with a history of fracture or surgery, respectively, Table 2). The insurance fee or area of residence was not strongly associated with “non-compliant”. Although statistically significant, the likelihood of “non-compliant” (OR) showed a minimal difference from 0.2% to 12% (Table 2).

## 5. Discussion

The objective of this study was to evaluate the difference in medication compliance among new users starting bisphosphonate according to the dosing interval. This study examined the MPR as an indirect marker of compliance with bisphosphonate therapy prescribed to large number of South Korean patients. We found that the MPR widely varied among various dosing intervals. The switch group and 3-monthly group showed the highest MPR, while the daily dose group showed the lowest MPR, indicating that long dosing intervals led to an increased compliance. The MPR was also affected by the patient’s inherent properties such as sex and age, so male patients and older patients aged over 80 tended to discontinue the drug more frequently. In contrast to expectation, a history of fracture or related surgery did not significantly increase the odds ratio of compliance. This reflected the relatively low awareness of the importance of anti-osteoporosis treatment, especially in particular groups such as older patients, males, and patients with a fracture history. The strength of the current study was that it utilized the data of 281,996 patients, using the national insurance database that covered almost all people in the country, thereby minimizing the risk of sampling bias. The prescription data and variables in our study was accurate because our national insurance agency payed costs based on the billing records of healthcare providers and all information was processed and collected electronically by a computer system.

Similar results were reported by other researchers [20,21]. Once-weekly dosing demonstrated consistently better MPR than daily dosing during a 1-year observation period in most studies [20,22,23,24,25]. The same trend was also reported in our study. However, even with weekly dosing, the compliance remained low, showing that 50% of the patients had less than 80% MPR, [22] and our data also showed a low compliance for weekly agents, i.e., 20% compliance for up to 2 years of observation; this value was not significantly different from the compliance value of the daily dose group (21%) in the current study.

The comparison of weekly versus monthly doses also showed an improved MPR with extended dosing intervals, [26,27] but the difference was not large and was occasionally controversial. USA military data showed that the overall rates of compliant patients (the ratio of MPR ≥ 80%) on weekly and monthly doses were 42.2% and 45.7%, respectively, for 1 year [28]. However, other researchers reported no significant differences in the effect of compliance or persistence on using administrative claims data, among new bisphosphonate users starting with weekly versus monthly dosing intervals [29,30,31]. In one study [31], a weekly dose (alendronate and risedronate) showed a 12-month median MPR of 61% and ibandronate (monthly or 3-monthly agent) showed a 12-month median MPR of 58% (in contrast, we reported weekly, monthly, and 3-monthly MPRs of 46%, 56%, and 65%, respectively in Figure 6). Briesacher et al. [29] reported a compliance rate of 49% for both dosing groups. We found much lower compliance rates of 20% and 28% for weekly and monthly doses than other studies. However, the monthly dose was associated with a significantly higher compliance rate. Many studies, including ours, supported the fact that compliance with the monthly dose was higher than that for the weekly dose [21,26,27,31].

Patients receiving a 6-monthly injection had a significantly higher persistence and compliance after 2 years than those receiving frequent doses (e.g., daily and weekly) [32]. This compliance for extended dosing intervals was also repeated in our data. Collectively, this evidence indicated that low-frequency dosing could significantly improve the level of compliance. Low-frequency dosing was proven to be related with good compliance in the systematic review of non-osteoporotic medication for other chronic conditions such as hypertension or cardiovascular disease [33]. The satisfaction with anti-osteoporosis medication was influenced by the side effects related to the drug [7], and this satisfaction was subsequently related with compliance. Low-frequency dosing was associated with few side effects and a high satisfaction. The monthly dose showed a similar efficacy in increasing BMD as a daily dose [9]. Extended intervals (3-monthly or yearly) also showed a similar or greater risk reduction in non-vertebral fractures with less inconvenience than daily or weekly intervals in a meta-analysis [10]. Thus, the extended dosing interval could be expected to increase the compliance without compromising the intended effect of fracture prevention.

We used MPR to estimate compliance in this study. Patients’ drug dosing histories could be measured by analyzing compliance (synonym: adherence) or persistence [15]. The two most common methods to measure a medication adherence were MPR and PDC (proportion of days covered), which were based on patient refill records [34]. These methods were secondary estimations compared to patients’ self-reports; however, they were more objective and adequate for administrative data. PDC is a newer and more conservative measure of adherence. MPR can overestimate medication adherence, as it counts the total number of days of medication supply, which may be increased by patients who obtain their prescriptions early [34]. We selected MPR as a marker for the adherence despite its shortcomings, because it was the most widely used method historically [35,36]. We thought that MPR would have an advantage in comparisons with the pre-existing data. The persistence, which was a slightly different concept with adherence, could be measured by looking at the permissible gap (for example, 180 days refill gap. It was widely used to study the continuation or discontinuation of drug treatment, although not as much as MPR [37,38].

The differences in MPR or compliance rates among various studies could be attributable to the differences in study designs and follow-up duration. Direct comparison was also complicated by differences in patient demographics. The inconsistency between studies could also be due to the prospective nature of some studies. In a prospective study, patients noticed that their medication-taking behavior was monitored, which resulted in increased medication persistence due to the so-called Hawthorne effect [39]. For example, in one randomized prospective study, researchers noted significant improvements in compliance with monthly agents (80.2%) as opposed to with weekly agents (73.3%), with a strikingly high overall compliance of over 70%, even with weekly agents [27]. We believed that our observational dataset with a large number of patients reflected real-world compliance. The compliance rate in the current study was much lower than that reported in other studies, especially the prospective one.

Identifying factors associated with reduced compliance is important to improve medication-taking behavior. It is still not clear which factors are strongly associated with poor compliance. In one study with multivariable models, several patient characteristics were independently associated with increased compliance: female sex, BMD testing before and after treatment, and fracture before and after treatment [6]. Factors that reduced the compliance included older age and comorbidities [6]. Another study also reported similar findings of a modest improvement in compliance with prior BMD test and reduced compliance with an increasing comorbidity index [29]. These findings, especially regarding to sex and age, were similar with our findings.

Regarding sex, we observed that the compliance was better among women than among men. In contrast to our findings, one study reported better compliance among male patients than among female patients; the odds ratio of compliance for female patients was 15% lower than that for male patients (adjusted OR = 0.851, 95% CI 0.788–0.932) [28]. However, another study showed that men had a 26% higher risk of discontinuing bisphosphonates therapy in comparison to women, which was a similar result as ours [38]. Generally, women were known to be less compliant with chronic medication than men [40]. However, in the case of osteoporosis, it was widely known that the awareness and treatment rates were lower among men than among women, with the treatment rate after diagnosis being 5.7% among men versus 22.8% among women based on the Korea National Health and Nutrition Examination Survey 2008~2011 [41]. Women were also more likely to receive osteoporosis treatment after a fracture than men [42,43,44,45]. The reason for the low prescription rate could be the low awareness of the seriousness of osteoporosis in men among both male patients and their treating physicians [46,47]. In our study, the absolute number of male patients was smaller than that of female patients; however, compliance among men was much lower than among women. Since mortality after hip fractures was higher in men than in women, the recommendations and appropriate education to continue with the prescription were required for the proper management of male patients [48]. One other possible reason for low MPR in our data was the differing severity of osteoporosis between men and women. In men, the severity of osteoporosis was not as severe as in women, so the possibility that the MPR was low could not be excluded because it was impossible to determine the severity of osteoporosis in our database.

The probability of taking an osteoporosis drug after fracture increased with age [43]; however, our study showed that compliance was much lower in older age. In group aged 60–69, the MPR was the highest at 52.7%, similar to the other study [6]. Especially in the presence of a fracture, the initial medication rate was higher in patients aged ≥80 than in younger patients, but the awareness of the importance of disease prevention appeared to decline rapidly over time, resulting in a corresponding decline in compliance [3]. In contrast to our study, in the GRAND 4 study conducted in Germany, younger patients under 60 years were significantly more likely to discontinue osteoporosis therapy than those aged ≥60 years. However, it was difficult to directly compare the two studies because the age intervals compared were different. Furthermore, other large-data studies also showed that the discontinuation of bisphosphonate increased among patients aged ≥80 years [38]. Patients aged over 80 most likely suffered from other chronic conditions preventing the intake of bisphosphonate, and they also had difficulty in traveling the distance to the outpatients’ clinic to obtain their prescriptions for osteoporosis. Families may also have believed that further treatment was not meaningful considering the patient’s age and accompanying underlying diseases.

We expected the compliance of patients with a history of fragility fracture to be higher than that of patients without such a history. However, our data were not consistent with the expectation or previous studies [6,38]. In one study, similar to our findings, the compliance of patients without fractures was better than that of patients with fractures [31]. Another study showed that the majority of the patients (94%) did not receive any osteoporosis medications within 1 year after a hip fracture and the MPR was 67% for combined weekly and monthly doses of bisphosphonate [49]. This finding indicated that patients with fractures were not as compliant as expected. However, there was the possibility that the recency of the fracture affected the discontinuation rate of the drug. Spanish data showed that recent fractures within 1 year decreased the probability to cease the bisphosphonates treatment (hazard ratio 0.92) in comparison with fractures occurring >1 year before [38]. In our study, fractures that occurred up to about 2 years ago were investigated, and the inaccuracy in determining whether a fracture was pathological or not seemed to cause these different results. Because the history of hip fracture increased the risk of subsequent fracture by 3.2-fold, these patients should be managed carefully [50,51]. This risk of repeated fracture was the highest in the first year of the index fracture. Because compliant patients with an MPR ≥ 0.8 showed a 14% lower risk of subsequent fracture than those with an MPR < 0.5, [31], it was important to increase compliance during the first year after the fracture.

One interesting finding of the current study was that the compliance of the “switch group” was the highest. Other studies also showed that patients who were exposed to other osteoporosis drugs before enrollment had a high compliance (83% for those with previous exposure vs. 74 % for those without), meaning that switching from other osteoporosis treatments could improve treatment compliance [52]. We did not investigate the reasons for this phenomenon, but the change to a new regimen in line with patient’s complaints or needs could help increase patient compliance. The patients in the switcher group would be the group of patients more committed to the treatment who decided to find a more tolerable dosing interval even in the case of side effects or the inconvenience of a previous regimen. However, our finding cannot be generalized easily because USA military data showed that switching drugs was related to a low compliance [28]. This issue should be explored in future studies.

There are several limitations to this study. First, owing to the relatively short follow-up duration, it was not easy to calculate the compliance of long dosing intervals, such as yearly dose. Yearly doses should have had very high MPR even if they were discontinued after the first dose in the setting of a short period of follow-up. Therefore, we excluded the data for yearly doses in this study. Currently, zoledronic acid 5 mg/100 mL is only agent injectable yearly; it is permitted as a treatment for Paget disease or osteoporosis in Korea. Second, we investigated the history of fracture and subsequent surgery, but whether the fractures were fragility fractures was not certain. We estimated that, in this specific age group, most fractures were induced by low-velocity slips/falls. High-velocity injury was present but at a low proportion; however, it did not affect the results. Another shortcoming is that we did not investigate the recency of fracture. Third, we only investigated compliance with bisphosphonate. Some patients may have transferred to other classes of drugs, such as the selective estrogen receptor modulator or denosumab (Prolia^®^), and if so, this did not mean that compliance had deteriorated. Therefore, it is necessary to study other agents simultaneously to obtain accurate statistics. Prolia^®^ or Forsteo^®^ (teriparatide) injections occupy a very small proportion of the Korean market, and most cases are not covered by national health insurance or partly covered for those who fulfill strict indications; therefore, we omitted these agents in the current study. Fourth, because our investigation was based on prescription claims, there was no way to estimate the real compliance of patients. We could not access the clinical data that may indicate when or why a patient discontinues or switches treatment. The reason why patients discontinue bisphosphonate therapy will vary, based on clinical reasons such as an inadequate response or adverse drug reactions. Fifth, selection bias may have occurred because we excluded the patients who died during investigation period. The percentage was as small as 2.6% for the total included patients, but resulted in the inaccurate estimation of compliance and could make an immortal time bias. Sixth, we could not obtain the information about the severity of the osteoporosis. The severity of osteoporosis could be one of the factors influencing the compliance and indication of medication. Lastly, we did not investigate comorbidity as a confounding factor, and this factor was important to determine the reason why some patients discontinued the drug, and would be an interesting topic in future studies.

## 6. Conclusions

Our findings showed that the initiation with a long dosing schedule could have a beneficial effect on patient compliance, independent of other non-modifiable factors, such as age, sex, or fracture history. However, while compliance with bisphosphonate therapy improved with longer dosing intervals, such as a 3-monthly dose, the overall compliance was still low indicating the need for strategies to improve compliance. Overall compliance was low among males, patients aged over 80, and not high in patients with a history of fracture. It is important to increase the compliance in these groups.

## Figures and Tables

**Figure 1 jcm-10-04350-f001:**
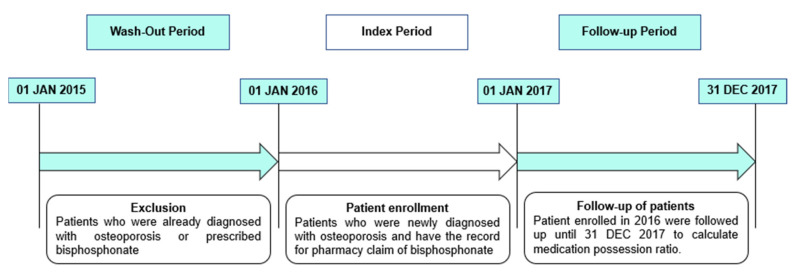
Description of study period: We recruited patients diagnosed with osteoporosis from January 2015 to December 2016. For these 2 years, the first 12 months were used as wash-out period and the osteoporotic patients in this period were excluded because we needed only newly diagnosed patients. We selected patients who were diagnosed with osteoporosis and prescribed bisphosphonates between 01 January 2016 and 31 December 2016. The first date of bisphosphonate prescription during this period was set as the index date. Only the patients whose data were available until 31 December 2017 were included.

**Figure 2 jcm-10-04350-f002:**
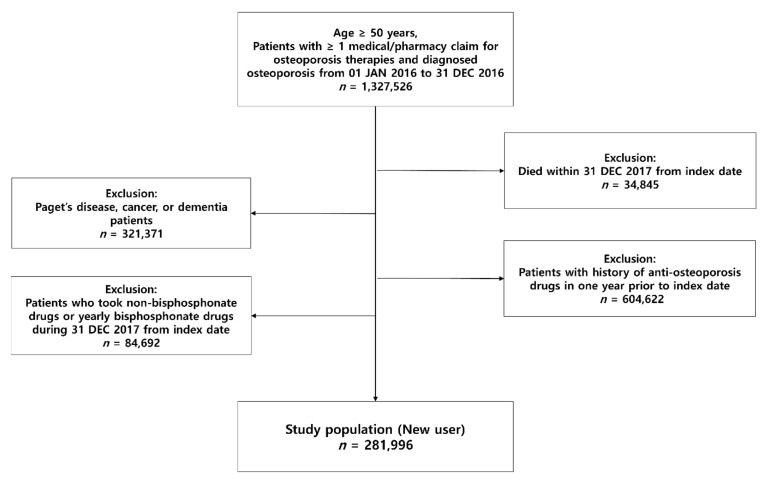
Study population: Initial enrollment and exclusion criteria. The first date of bisphosphonate prescription during this period was set as the index date.

**Figure 3 jcm-10-04350-f003:**
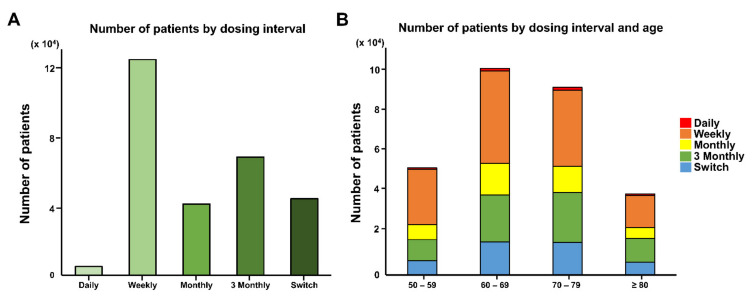
Distribution of dosing interval: During the study period, weekly bisphosphonate was the most prescribed dose, and the daily dose was the least prescribed one (**A**). Number of patient and prescription pattern according to patient’s age were shown. The number of osteoporotic patients with bisphosphonate prescription was highest in the ages 60-69. The analysis of the prescription pattern showed a similar trend for all ages (**B**).

**Figure 4 jcm-10-04350-f004:**
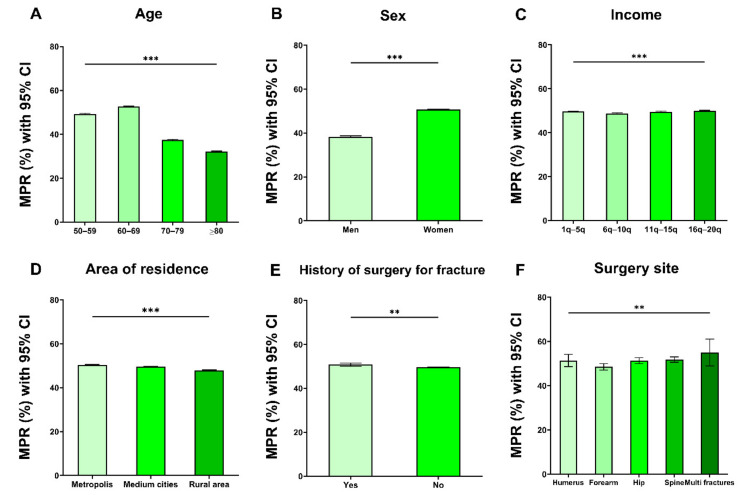
Drug compliance according to the patient’s demographic factors: Drug compliance according to age interval (**A**), sex (**B**), income (**C**), area of residence (**D**), presence of fracture (**E**), and fracture site (**F**). The bar is 95% confidence interval. <0.01 **, <0.001 ***.

**Figure 5 jcm-10-04350-f005:**
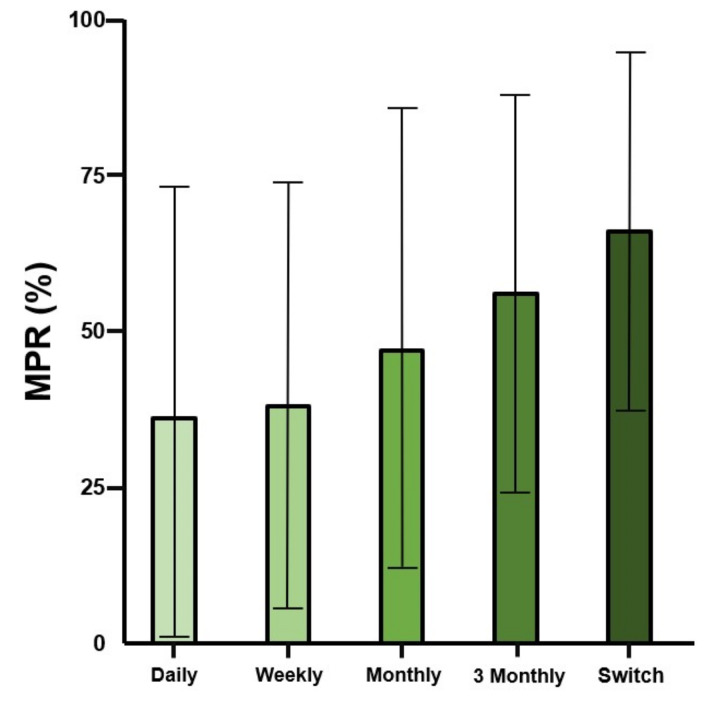
Drug compliance as medication possession ratio (MPR) according to dosing interval: As expected, extended dose groups had high drug compliance. The switch group had the highest drug compliance. Standard deviation is shown as range in the graph.

**Figure 6 jcm-10-04350-f006:**
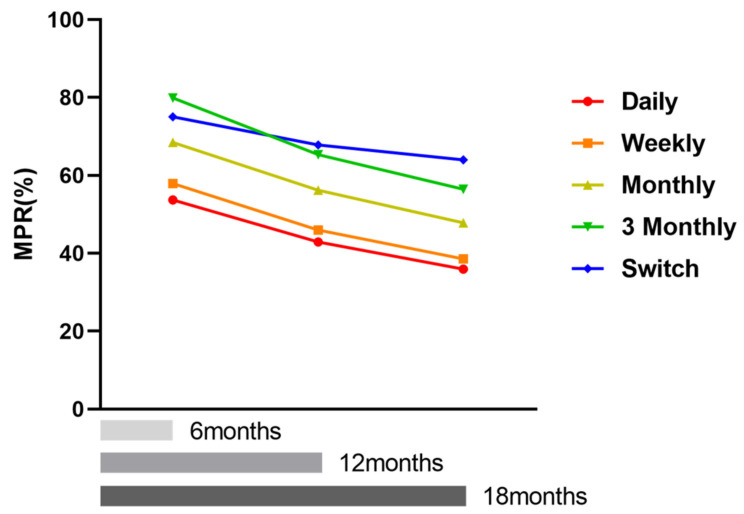
Drug compliance as MPR by each follow-up duration and dosing interval: Drug compliance gradually decreased regardless of dosing interval after prescription in all groups.

**Figure 7 jcm-10-04350-f007:**
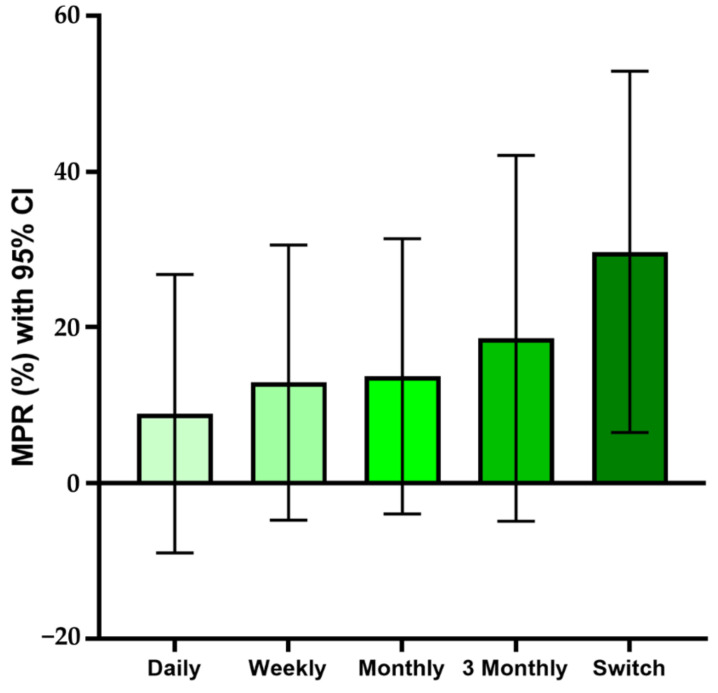
Drug compliance as a medication possession ratio (MPR) according to dosing interval, longer-term investigation: Overall MPR was lower than with shorter-term investigations. Longer dosing interval groups had higher drug compliance. Confidence interval is shown as range in the graph.

**Table 1 jcm-10-04350-t001:** Univariate comparison of compliance (MPR ≥ 80%) according to patients’ demographic subgroup.

Variable		Noncompliance (MPR < 80)	Compliance (MPR ≥ 80)	*p*-Value *
Sex	Men, *n* (%)	20,218 (9.65)	4573 (5.53)	<0.001
	Women, *n* (%)	189,205 (90.35)	78,156 (94.47)	
Age	50–59, *n* (%)	38,595 (18.43)	14,613 (17.66)	<0.001
	60–69, *n* (%)	70,608 (33.72)	32,243 (38.97)	
	70–79, *n* (%)	67,436 (32.2)	26,823 (32.42)	
	≥80, *n* (%)	32,784 (15.65)	9050 (10.94)	
Insurance fee	1–5q	57,599 (27.5)	22,767 (27.52)	<0.001
	6–10q	31,751 (15.16)	12,022 (14.53)	
	11–15q	44,891 (21.44)	17,614 (21.29)	
	16–20q	75,182 (35.9)	30,326 (36.66)	
Area of residence	Metropolis	81,012 (38.68)	33,054 (39.95)	<0.001
	City	94,760 (45.25)	37,410 (45.22)	
	Rural area	33,648 (16.07)	12,263 (14.82)	
Dosing interval	Daily, *n* (%)	3445 (1.70)	922 (1.17)	<0.001
	Weekly, *n* (%)	98,431 (48.51)	24,974 (31.57)	
	Monthly, *n* (%)	29,365 (14.47)	11,513 (14.56)	
	3 Monthly, *n* (%)	45,063 (22.21)	23,512 (29.73)	
	Switch, *n* (%)	26,594 (13.11)	18,177 (22.98)	
Fracture history	No, *n* (%)	193,661 (92.47)	76,122 (92.01)	<0.001
	Yes, *n* (%)	15,762 (7.53)	6607 (7.99)	
Surgery history	No, *n* (%)	203,474 (97.16)	80,331 (97.1)	0.4038
	Yes, *n* (%)	5949 (2.84)	2398 (2.9)	

* Chi-square test.

**Table 2 jcm-10-04350-t002:** Analysis of multivariate logistic regression of being non-compliant (MPR < 80%) according to various predefined demographic factors and dosing interval.

Variable		Odd Ratio (95% CI) *	*p*-Value
Sex	Male vs. Female	1.389 (1.342, 1.439)	<0.001
Age	50–59 vs. 80+	0.645 (0.625, 0.665)	<0.001
	60–69 vs. 80+	0.565 (0.549, 0.581)	<0.001
	70–79 vs. 80+	0.674 (0.655, 0.693)	<0.001
Insurance fee	2qu vs. 1qu	1.051 (1.023, 1.079)	<0.001
	3qu vs. 1qu	1.024 (1, 1.048)	0.0539
	4qu vs. 1qu	0.966 (0.946, 0.987)	0.0012
Area of residence	Small city vs. Rural area	0.909 (0.887, 0.931)	<0.001
	Metropolis vs. Rural area	0.877 (0.855, 0.899)	<0.001
Dosing interval	Weekly vs. Daily	1.1 (1.02, 1.184)	0.0122
	Monthly vs. Daily	0.737 (0.682, 0.795)	<0.001
	3 Monthly vs. Daily	0.54 (0.501, 0.582)	<0.001
	Switch vs. Daily	0.415 (0.385, 0.447)	<0.001
Fracture history	Yes vs. No	0.885 (0.856, 0.914)	<0.001
Surgery history	Yes vs. No	0.945 (0.897, 0.996)	0.0334

* Adjusted for all factors in this table.

## Data Availability

The datasets used and/or analyzed during the current study are available from the corresponding author on reasonable request.

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
