# Peer review of "Effect of Dosing Interval on Compliance of Osteoporosis Patients on Bisphosphonate Therapy: Observational Study Using Nationwide Insurance Claims Data"

_jcm, 2021, doi:10.3390/jcm10194350_

Round 1

Reviewer 1 Report

Accept. No further comments.

Author Response

Thank you for your kind review.

Reviewer 2 Report

Dear Editor, 

The present manuscript has two major problems 1- most demographic data were associated with the pattern of dosing interval without any medical indication.

Moreover, the MPR index indicating the frequency of drug administration following the observation period was correlated with dosage interval and other parameters without considering the diagnosis, grade of osteoporosis,  etc. 

Author Response

  1. Most demographic data were associated with the pattern of dosing interval without any medical indication.

: Unfortunately, in our study design, it was not possible to recognize actual medical indications in each patient. We assumed that the most prevalent indication for bisphosphonates was osteoporosis in this population. We also excluded the patients who had the underlying disease, which could affect drug prescription such as cancer, dementia, or Paget’s disease. And this was indicated in the manuscript.

  1. Moreover, the MPR index indicating the frequency of drug administration following the observation period was correlated with dosage interval and other parameters without considering the diagnosis, grade of osteoporosis, etc.

: I agree with your opinion, therefore we added this issue as one of the limitations. In our data structures, it was impossible to check the severity of osteoporosis because BMD data was not available. We believe that all included patients have been diagnosed with osteoporosis after BMD since it is a routine practice in our country, however, it is still expected that the severity of osteoporosis will vary among patients.

: "Sixth, we could not obtain the information about the severity of the osteoporosis. The severity of osteoporosis could be one of the factors influencing the compliance and indication of medication"

Reviewer 3 Report

This study presents valuable observational data about compliance to bisphosphonates on a very large population. The design is appropriate. The study is mainly confirmatory as low compliance to bisphosphonates was already known. However, I feel this study can provide important information especially relative to the South Korean population.

There are minor concerns that need to be addressed to improve this manuscript:

  • please specify why the type of bisphosphonate was not available (alendronate, risedronate, etc, ...). If this information is not available, that should be mentioned in the limitations
  • please specify which bisphosphonate is usually given every three months in this population
  • please rephrase  'the p-value does not make sense' with another wording, in the statistical analysis section
  • please present tables in order (supplementary tables are usually shown at the end)
  • I would avoid both figures on page 9, they are a bit confusing as they are very similar to the other preceding figures. Please delete them and state in the text that you used the same methods as those used in the main group analysis.

Author Response

Thank you for your valuable review.

please specify why the type of bisphosphonate was not available (alendronate, risedronate, etc, ...). If this information is not available, that should be mentioned in the limitations

: We included this information as a supplementary table 1.

please specify which bisphosphonate is usually given every three months in this population, please

: We included this information as a supplementary table 1.

rephrase 'the p-value does not make sense' with another wording, in the statistical analysis section

: We changed it.

"The p-value has limited value in large-scale data"

please present tables in order (supplementary tables are usually shown at the end)

: We move supplementary figure and table to the end of the manuscript

I would avoid both figures on page 9, they are a bit confusing as they are very similar to the other preceding figures. Please delete them and state in the text that you used the same methods as those used in the main group analysis.

: I agree with your point. I believe that this problem can be solved by moving the supplementary figure to the end of the manuscript.

Round 2

Reviewer 2 Report

Dear Editor 

the authors changed sufficiently the manuscript according to my suggestions 

I agree to review

This manuscript is a resubmission of an earlier submission. The following is a list of the peer review reports and author responses from that submission.

Round 1

Reviewer 1 Report

Peer review

Title: Effect of Dosing Interval on Compliance of Osteoporosis Pa- tients on Bisphosphonate therapy: Observational Study using Nationwide Insurance Claims Data

The paper report the results of a observational study assessing the medication possession ration as a measure of the compliance with bisphosphonates treatment. The authors utilized a Korean claims database where most population is included. The effect of the age, sex, dosing interval, surgery of fractures, among other on the compliance was evaluated. The paper focus mainly on the increase in the MPR with the increase in the dosing interval.

Results:

Figure 2. Really informative and nice Figure. Please clarify what is the 'index date' any part in the Figure

Figure 3. Please notice that Figure 3 provides information already presented in Table 1. Consider remove any of them. Please avoid repeated information.

Table 2:

  • Please clarify whether % are over total in columns or total in rows. Thank you
  • Please clarify: the sum of proportions by columns does not sum up to 100%. Please review and correct, otherwise, if proportions must not sum up to the total by columns, provide an explanation of the denominator utilized.
  • Please , in footnote, inform whether ‘surgery’ and ‘fracture’ occurred during index period

Figure 4: Similar to my comment to Figure 3. Please notice that Figure 4 provides information already presented in Table 2. Consider remove any of them. Please avoid repeated information along the paper.

Figure 5:

  • Please consider the following title: Drug compliance as medication possession ratio (MPR) according to dosing interval .

  • In that case, please delete the following sentence: '...Compliance is shown as MPR'

  • Please clarify what range is reported in figure, is it 95% confidence interval?

Figure 6:

  • Please explain why switching is not reported in this Figure 6.
  • Please consider the following title: Drug compliance as medication possession ratio (MPR) by each follow-up duration and dosing interval

For all figures: Please avoid repeated in formation in figures' headings and Figures' title.

Table 4:

  • Please consider adding '(MPR ≥ 80%) ' after '...being compliant' as follows in title: '...being compliant (MPR ≥ 80%) ...'
  • If 'drug frequency' and 'dosing interval pattern' refer to the same thing, i.e. the same definition, You may want to unify both terms and use only one of them in order to avoid confusion to the reader.
  • Please unify ‘operation’ and ‘surgery’ if you refer to the same thing. Thank you
  • Please be more precise when mentioning 'chronic disease', what chronic disease did you adjusted for?
  • Please justify why OR are not adjusted by all factors reported in Table 4 (i.e. insurance fee, area of residence.
  • Also, Please provide the information on chronic disease in Table 4.

Methods:

Please clarify if ‘operation’ means ‘surgery for pathologic fracture’ as stated above. Please unify. Also please clarify in the tables.

Please clarify if switching means that the ‘drug’ itself (the active substance) was changed or the dosing was change or both.

Discussion:

After adjusting by all variables resulted associated in the OR model (at least all those reported in Table 4), please mention all factors you found associated with the compliance such as chronic disease, operation, residence or insurance fee, etc. and discuss.

Please review the mention to 'high MPR and ' in this sentence. Is it correct or you refer to 'This compliance for extended dosing intervals was also repeated in our data '

Please review 'for the other chronic condition' , clarify what is it?

It seems that the authors intend to say that prospective studies do not compensate for that shortcoming. Please clarify your point.

Please consider comparing your results with the following reference as well, in which men had a 26% higher risk to discontinue bisphosphonates therapy in comparison to women and discontinuation increased among the patients aged >=80 years.:

Elisa Martín-Merino et al. Cessation rate of anti-osteoporosis treatments and risk factors in Spanish primary care settings: a population-based cohort analysis Arch Osteoporos. 2017 Dec;12(1):39. doi: 10.1007/s11657-017-0331-6. Epub 2017 Apr 11

Please clarify what do you refer to by 'treatment rate' here. 5.7% and 22.8% seems to be really low, please declare the time those percentages are calculated for.

Sex: Thus it can be that those men treated with bisphosphonate had a different indication (less severe or better prognosis) than women treated. Please discuss: could your data (i.e. higher compliance among men than women) be confounded/explained by the prognosis or severity of the indication?

According to OR reported in table, the age categories determine up-to 44% reduction in compliance (OR: 0.56) similar to the OR reporter for dosing intervals (OR: 0.54). If that is correct, please consider to modify the following sentence that may not be correct 'The difference in compliance with age was not as large as compared to the difference in compliance with dosing intervals. '

Please clarify the sentence 'importance of disease prevention decreases rapidly over time' Do you mean that the effect of the bisphosphonates decreases over time?

Please, discus this a little further, for instance discus factors that could determine the discontinuation in younger versus older, maybe indication, perception of prognosis etc.

Please consider comparing your results with the following reference as well, in which recent fractures within 1 year decreased the probability to cease the bisphosphonates treatment (0.92; 95% CI: 0.88 0.96)  in comparison with fractures occurring >1 year before. The recency of the fracture can be discussed in your paper as a reason to determine the compliance:

Elisa Martín-Merino et al. Cessation rate of anti-osteoporosis treatments and risk factors in Spanish primary care settings: a population-based cohort analysis Arch Osteoporos. 2017 Dec;12(1):39. doi: 10.1007/s11657-017-0331-6. Epub 2017 Apr 11

Please discuss any reason. For instance , Could it be that switcher were patients that communicated their physicians about experienced side effects or feeling uncomfortable regimens?. i.e. their could be patients more committed with the treatment that decide to find a tolerable treatment and thus communicate medications problems/uncertainties etc. with the prescription. I do not know

Could you please mention what bisphosphonates are available yearly? for what indication? thank you. It can be informative for the reader.

Also, in the following sentence of Discussion:

 ‘Yearly doses should have very high MPR even if they are discontinued after the first dose. ’ Do you mean that MPR is not a good estimation of compliance for measuring the yearly compliance? please clarify

Maybe, the limitation is that you could not measure recent fractures (rather than the type of fractures frailty/not)? please consider mentioning it.

Please clarify the statement that ‘…osteopenia; therefore, the analysis of this medication was difficult to perform ’, why should osteopenia be difficult to perform?

Conclusions

It is important to mention the factors adjusted for in that conclusion and highlight that it is a modifiable factor while the other factors analyzed are not. As an example: ‘Our findings showed that initiation with a long dosing schedule can have a beneficial effect on patient compliance independently of other non-modifiable factors such as age, sex, ¿having a? chronic disease, fracture history or ¿any recent? operation.

Please clarify for instance: In order to get effective treatments it is important to increase compliance in these groups’

Reviewer 2 Report

Hyun Il Lee and colleges investigated compliance of Bisphosphonate using Korean nationwide insurance claims data. In general, the study is well described and well written. Overall, the findings are very descriptive, but this fit the aim of getting more insight in compliance among dosing interval stratified on various groups.

Detailed review is found below, point 0-29.

Unfortunately, I see some potential problem in the study design, that may affect the findings.

I) Either only patients being ‘alive’ and possible to follow during the full study period are included and followed, or patients are handled as if they are alive during the study period. If a more precise picture of compliance is wanted, then patient should be followed from study start and until death, migration, deprescribing, switch of drug, end-of-study etc. Similar to a survival analysis. I cannot figure out the consequences of the current design, but there is some selection bias, and potentially also immortal time bias.

II) Compliance is defined as MPR, calculated as duration of prescription divided in duration of observation period. This is simple, and intuitive. It is an assumption that if patient are getting a prescription, they also take the medication. This cannot be more precise in the available data. However, related to I) above, I am not sure that the observation period is truly calculated.

III) Due to I) and II) above, and the relative short study period, the findings of better compliance among switchers and among the longer dosing interval is due to the study design. The long-term dosing intervals may have the best ‘starting point’. Also, it may be so, that if MPR for the switching group is calculated at each starting dose, when compliance will look good/high.

At least the limitation and possible impact of these potential  biases must be included in the manuscript and discussed.

Detailed review. Headlines are marked with ** ***.

**Author Contributions. **

0) I believe that the description under Auhtor Contribution does not allow all authors to be coauthors according to ICMJE guidelines http://www.icmje.org/recommendations/browse/roles-and-responsibilities/defining-the-role-of-authors-and-contributors.html

**introduction**

1) The following sentence/statement need a reference: “ Bisphosphonates account for about 70% of all anti-osteoporosis drugs.”

2) Fine to mention your hypothesis in the introduction. However, if it should be really useful, and to make transparency, you should include explicit the direction of the compliance vs. dosing interval. E.g. longer dosing interval, higher/better compliance.

3) I think the following sentence is not correct/as intended: “This study investigated the dose interval associated with the highest drug compliance and the patient-related factors affecting the compliance”. Maybe the authors meant: “This study investigated IF [added] the dose interval WAS [added] associated with [deleted: the highest] drug compliance and the patient-related factors affecting the compliance”

** Materials and Methods**

***Data source***

1) In general, I suggest to avoid using the term retrospective in describing study design, as retrospective can be used and is understood very different. A suggestion to the sentence, and how I believe that the authors meant was: “We conducted an observational study using administrative claims data in South Korea. Administrative claims data initially collected for  issuance purpose, but is highly valuable for research”. Or simple, the authors can delete the word retrospective. The data origin is well described in the following sentences.

2) suggest change of “population of our country” to “population of Korea”

3) I believe that a study cannot be performed in accordance with STROBE guidelines, but can be reported in accordance with STROBE guidelines?

***study population***

4a) The authors write “We included only patients whose data were available until December 31, 2017 for a minimum of 1-year follow-up.”. It sounds like patients needed to be available at DEC 2017? The text to Figure 1 also indicate that only patients with available data until December 2017 were included. If so, immortal time bias is a problem. Patients starting drug treatment in 2016, but not surviving until DEC 2017 will not be included. Surviving is to be understood as being a live (in contract to being dead), but can also be ‘present in data’, e.g. due to migration, change of health care insurance not covered in the NHIS database. This may lead to either over or underestimation of compliance, and may affect the identification of factors affecting compliance. Either survival analyses with censoring should be use, or at least the potential limitation should be discussed in the Discussion section.

4b) partly related to 4a) above, the switching group is a potential black box. If patients are followed from initiation until treatment end, death, censoring or end of study, the switching group could be split up, so that the patients will contribution to the relevant group (e.g. weekly and monthly) with the periods where they exposed. E.g. if 6 months on weekly, the MPR can be calculated for these 6 months, and then e.g. 6 months of monthly. Maybe this can be done, even in the current design where patient are following until DEC2017 without censoring, deprescribing, death, etc. taken into account.

5) the wash out is fine, but the authors write: “used as wash-out data to exclude the patients who had ever been pre-scribed bisphosphonates before”. If the authors agree, I suggest it should be “used as wash-out data to exclude the patients who had [deleted: ever] been pre-scribed bisphosphonates before [added: the index period]”

6) excluding patients who die also lead to potential bias, that may lead to either over or underestimation of compliance, and may affect the identification of factors affecting compliance. This is related to point 4). Either survival analyses with censoring should be use, or at least the potential limitation should be discussed in the Discussion section.

7) Figure 1: I believe it should not be “index-date period”, but just “index period”. The box of ‘follow-up of patients. The end date should be 12 and not 13, I believe. In general: Using a date format such as 31DEC2021 is better as this can not be misunderstood.

8) should figure 2 be part of the result section?

9) based on text, figure 1 and figure 2: it is not clear to me, if the FIRST 12 months, from JAN2015 to DEC2016 are used as washout (as indicated in the text for figure 1). It should be the same ‘relative’ period for ALL patients. E.g. patients exposed (having index) in e.g. FEB2016 and patients exposed in NOV2016, should use a period as close to the index, and of the similar duration, e.g. 12 months, being from FEB2015 and NOV2015, respectively. Text in figure 2 state “history of anti-osteoporosis drugs in one year prior to index date” that sounds like a correct way to do it. What has been done? This needs to be clearly described, ideally one place in the text.

***measure of compliance***

10) The authors write: “Patients’ drug dosing histories can be measured by analyzing compliance (syno-nym: adherence) or persistence, but we only measured compliance in this study. To min-imize the confusion with similar terms,” I believe compliance, adherence and persistence are NOT similar terms or synonyms, although overall this is about drugs and how the drugs are taken – they cover the subject slightly different. I suggest that the authors write something like: “We define compliance according to the Interna-tional Society for Pharmacoeconomic and Outcomes Research as […]”

11) regarding calculation of medication possession ratio (MPR). Compliance is defined as agreement between observation period and duration of prescribed medication. What if the patient was deprescribed in the period? Then the denominator should be less than the ‘duration of follow-up’. It should be days between index date and date of deprescribing. This may increase compliance/the MPR.

12) The authors have no reference to definition of MPR, or the criteria/cut-off of 80%. I presume there is a lot of literature discussing definitions of compliance - The authors need to elaborate on the method used. At least in the discussion.

***statistical analyses***

13) Fine to write that p-value is not determine if a covariate is deemed important for compliance. That is the current movement away from blind trust in p-values. The authors state that “emphasis was placed on the difference between groups or the odds ra-tios of the groups”. What about confidence interval? Wide or narrow confidence interval are important to take into account, irrespectively of the p-value.

**Results**

14) Table1: It is my understanding that the percentages are summarized to 100 horizontally. However, for insurance fee 1q-5q the sum of percentage of the treatment groups is 56.85 and not 100. The table should be checked.

15) The sentence “The analysis of the prescription pattern in terms of dosing interval showed a similar trend in all age groups (Figure 3B).” I suggest to reformulate to something similar to: “Figure 3B show the dosing interval, stratified on age group. A similar trend was seen in all age groups.” The problem with the current sentence is that it mention an analysis, that is truly a figure, and that “the prescription pattern in terms of dosing interval” is a complex formulation.

16) It is my understanding that the study have information about prescriptions, and not patient preferences per se. Hence, the sentence “Patients from rural areas showed more preference for” should be just “Patients from rural areas were most likely to receive”

17) The following sentence is not needed in the Result section. Furthermore, no p-values (or 95%CI) are presented, therefore I suggest to delete: “The difference in dosing interval according to demo-graphic factors was statistically significant because of the extremely large number of pa-tients. Therefore, we described only the observation that we consider meaningful.”

***drug compliance****

18) Table 2: with SD provided, it is difficult to interpreted difference between groups. Is the reader to compare across groups and dosing interval? Then the SE or 95%CI are needed.

19) figure 4: what show the whiskers?

20) when the authors write: “Mean MPR was higher in women significantly (50.7% versus 38.3% for men, Figure 4B).” it is not clear where this is seen? There is no test or comparison of point estimate and 95%confidence interval for women and men?

21) Similar, for the other comparison across groups with reference to the panels of figure 4: it is not clear how the authors can state statistically significance/insignificance.

22) Figure 5: what show the whiskers?

23) Figure 6: what is the blue line? Not in the legend. Where is the yellow line for the monthly dosing interval group?

24) it is found that highest MPR is seen among switchers and among the longer dosing interval. It is my concern that this is observed, due to the study design: since compliance is calculated as duration of prescription divided by observation time (without any right censoring as commented above), the long-term dosing intervals are having the best starting point. Also, it may be so, that if MPR for the switching group is calculated at each starting dose, when compliance will look good/high.

***factors affecting compliance rate***

25) as with 24) above, the study design may affect the findings in the analyses of below/above 80% MPR.

**Discussion**

26) As the study strength of the study, only population size is mentioned. Do the authors not trust the measurements/variables in the databases? The precision of the information of prescriptions? What about the study design?

27) the authors write: “Although the weakness of our data set is its retrospective nature, a large number of patients can compensate for that shortcoming, so we believe that this study reflects real-word com-pliance.”. I believe this should be reformulated. First, I suggest to not describe the data as retrospective. This can be (mis)understood as if the data is collected back in time, e.g. by asking patients leading to potential recall bias. That is not the case in the present study. Rather, the data should be described as data from (claim) databases, and collected not for research purpose, but very useful for this since it reflects real world. However, there may be some mis-coding/errors in data. Next, a big population cannot counter data with errors. A lot of ‘bad data’ is not better. So I suggest to remove this, unless the authors can explain how they utilize the (big) population size by e.g. imputation, regressions, etc.

28) among limitations to the study: the authors mention that yearly dose will by definition look like a high compliance. That is my point for also the other groups with longer dosing interval, mentioned in point 24 above.

29) among limitations to the study: the authors mention other drugs, that was not investigated. When investigating bisphosphonate, stop of bisphosphonate due to start on other drugs should be taken into account to limit the duration of follow-up. This is related to point 11 above.